# Production, Characterization, and Industrial Application of Pectinase Enzyme Isolated from Fungal Strains

**Sudeep KC [1,2], Jitendra Upadhyaya [3], Dev Raj Joshi [2], Binod Lekhak [2],**
**Dhiraj Kumar Chaudhary [4], Bhoj Raj Pant [5], Tirtha Raj Bajgai [6], Rajiv Dhital [7],**
**Santosh Khanal [8], Niranjan Koirala [9,10,*] and Vijaya Raghavan [3,\*]**

[1] Department of Public Health/Pharmacy, Central Institute of Science and Technology, Pokhara University, Kathmandu 44600, Nepal; sudeep0137@gmail.com

[2] Central Department of Microbiology, Tribhuvan University, Kathmandu 44600, Nepal; joshi_devraj@hotmail.com (D.R.J.); binod.lekhak@microbiotu.edu.np (B.L.)

[3] Department of Bioresource Engineering, McGill University, Macdonald Campus, 21, 111 Lakeshore Road, Ste-Anne-de-Bellevue, Montreal, QC H9X 3V9, Canada; jitendra.upadhyaya@mcgill.ca

[4] Department of Life Science, College of Natural Sciences, Kyonggi University, Suwon 16222, Korea; dhirajchaudhary2042@gmail.com

[5] Nepal Academy of Science and Technology, Khumaltar, Lalitpur 44600, Nepal; bhojraj.pant@nast.gov.np

[6] Minhas Microbrewery, Distillery and Winery, 1314 44 Ave NE, Calgary, AB T2E 6L6, Canada; tirraj@yahoo.com

[7] Department of Food Science, University of Missouri, Columbia, MI 65211, USA; dhitalrajiv7@gmail.com

[8] Department of Microbiology, National College, NIST, Khusibu, Kathmandu 44600, Nepal; santoshkhanal007@gmail.com

[9] Department of Natural Products Research, Dr. Koirala Research Institute for Biotechnology and Biodiversity, Kathmandu 44600, Nepal

[10] Laboratory of Biotechnology, Faculty of Science and Technology, University of Macau, Macau SAR 999078, China

\* Correspondence: koirala.biochem@gmail.com (N.K.); vijaya.raghavan@mcgill.ca (V.R.)

**Abstract:** Pectinases are the group of enzymes that catalyze the degradation of pectic substances. It has wide applications in food industries for the production and clarification of wines and juices. The aim of this study was to isolate, screen and characterize pectinase from fungi isolated from various soil samples and evaluate its application in juice clarification. Fungal strains were isolated and screened primarily using 1% citruspectin incorporated potato dextrose agar (PDA) and secondarily using pectinase screening agar medium (PSAM) for pectinolytic organisms. The enzyme was produced by submerged state fermentation and assayed using the dinitro salicylic acid (DNS) method. From 20 different soil samples, 55 fungal isolates were screened primarily and, among them, only 14 isolates were subjected for secondary screening. Out of 14, only four strains showed the highest pectinolytic activity. Among four strains, *Aspergillus* spp. Gm showed the highest enzyme production at a 48-h incubation period, 1% substrate concentration, and 30 °C temperature. The thermal stability assessment resulted that the activity of pectinase enzyme declines by 50% within 10 min of heating at 60 °C. The optimum temperature, pH, and substrate concentration for the activity of enzyme was 30 °C (75.4 U/mL), 5.8 (72.3 U/mL), and 0.5% (112.0 U/mL), respectively. Furthermore, the yield of the orange juice, the total soluble solid (TSS), and clarity (% transmittance) was increased as the concentration of the pectinase increased, indicating its potential use in juice processing. Overall, the strain *Aspergillus* spp. Gm was identified as a potent strain for pectinase production in commercial scale.

**Keywords:** pectinase; *Aspergillus* spp.; DNS assay; fungal isolate; pectinolytic activity

---

## 1. Introduction

Pectin is an important component of the middle lamella and primary cell wall of higher plants. Pectins are high molecular weight acidic heteropolysaccharide primarily made up of $\alpha$ (1−4) linked D-galacturonic acid residues [1]. Three major pectic polysaccharides groups are recognized, all containing D-galacturonic acid to a greater or a lesser extent. They are homogalacturonan (HG), rhamnogalacturonan I (RGI), and rhamnogalacturonan II (RGII) [2–4].

Pectinases are a group of enzymes that degrade pectic substance and are classified according to their mechanism of action. For example, methylesterases remove methoxy groups from highly or partially esterified galacturonan. Polygalacturonases catalyze the hydrolysis of the glycosidic bonds in a random fashion (endopolygalacturonase) or from the nonreducing end of homogalacturonan releasing galacturonic or digalacturunic acid residues (exopoly-glacturonases) [5,6]. Pectinolytic enzymes, or pectinases, are also classified according to their mode of action and their substrate: polygalacturonases, which are sub classified as endo-polygalacturonases (E.C. 3.2.1.15) and exo-polygalacturonases (E.C. 3.2.1.67); lyases, which are sub classified into pectatelyases (E.C. 4.2.2.9 and EC. 4.2.2.2) or pectin lyases (E.C. 4.2.2.10); and pectin methylesterases (E.C. 3.1.1.11). It is recommended to use a combination of different kinds of pectinases, along with other enzymes such as cellulases and hemicelullases, as multiple enzymes can degrade different parts of the polymer, resulting in the maximal degradation of the pectin in various raw materials such as in citrus juice processing [7,8]. Studies have reported that pectinase of microbial origin accounts for 25% of global food and industrial enzymes sale and their market is increasing continuously [9]. Additionally, enzymes comprise a well-established global market projected to reach USD 6.3 billion in 2021 [7]. Microorganisms including fungi are promising sources of enzymes. Fungi produces numerous extracellular enzymes that possess a special effect in the decomposition of organic matter. These include pectinolytic enzymes which are excreted to break down the middle lamella in plants so that it can insert fungal hyphae and extract nutrients from the plant [10,11]. In addition to fungi, pectinolytic enzymes are naturally produced by many other organisms like bacteria, insects, nematodes, and protozoans [12]. For the commercial production of pectinases, *Aspergillus* spp., *Erwinia* spp., *Bacillus* spp., and *Penicillium* spp. have been extensively used [9,13].

Pectinases have crucial roles in food industries. These enzymes are useful for fruit juice extraction and wine clarification; tea, cocoa, and coffee concentration and fermentation; vegetable oil extraction; preparation of jam and jellies; and pickling [14,15]. Furthermore, these enzymes are used in paper and pulp industries, bleaching of paper, bio-scouring of cotton, retting and degumming of plant fibers, oil extraction, wastewater treatment, poultry feed additives, protoplast fusion technology, and bioenergy production [10,15].

Enzyme breakdown of the biomolecules depends upon the type of microorganisms, fermentation condition such as pH, incubation time or cultivation time, carbon and nitrogen source, types and concentration of substrate, temperature, agitation, and use of different enzyme preparations [16,17]. The application of new enzymes with desirable biochemical and physicochemical characteristics and low-cost production in commercial processes has always been regarded as essential research. Keeping all the advantages into consideration, the objectives of this study were (1) to isolate and screen pectinase-producing fungi from the soil samples, (2) to optimize different parameters for maximum enzyme production and evaluate the enzyme activity with various parameters, and (3) to evaluate its potentiality in juice clarification.

## 2. Materials and Methods

### 2.1. Soil Sampling

Soil samples were collected from various regions of Nepal, geographically located at Gulmi (GPS location: 28°03′60.00″ N 83°14′60.00″ E), Manang (GPS location: 28°33′7″ N84°14′27″ E), and Kathmandu (GPS location: 27°42′6.08″ N 85°19′14.16″ E). These sampling sites varied in altitude

ranging from 1700 to 5416 m and were rich in spoiled fruits, agro-industrial wastes, fruit pulp, composts, decaying leaves, and organic fertilizers. Altogether, 20 soil samples were collected from the depth of 20 cm into sterile zip-lock bags and transported to the laboratory for analysis.

*2.2. Isolation, Screening, and Identification of Pectinolytic Fungi*

Potato dextrose agar (PDA, Himedia) incorporated with 1% citrus pectin was used for the isolation and screening of pectinolytic fungi from the soil samples. For isolation, a 5 gm soil sample was transferred to 45 mL of distilled water ($10^{-1}$) serially diluted up to $10^{-6}$. From dilutions, 0.1 mL was inoculated by spread plate method in the prepared agar plates and incubated at 25 °C for 3–5 days. Different colonies were selected and subcultured onto the two different PDA plates containing pectin for duplication and incubated at 25 °C for 3–5 days. Fungal colonies with distinct morphology were selected and subcultured repeatedly to obtain pure culture by point inoculation. For primary screening, 1% (*w/v*) cetyltrimethylammonium bromide (CTAB) was flooded on one agar plate containing selected fungal colonies and incubated at 25 °C for 1 h. The zone of hydrolysis around the colonies indicated the pectinolytic activity of fungi and the colonies were preserved for further study. The primary screened fungal isolates were inoculated on pectinase screening agar medium (PSAM) and incubated at 25 °C for 42 h. The PSAM contains (in g $L^{-1}$): $(NH_4)_2HPO_4$, 3.0; $KH_2PO_4$, 2.0; $K_2HPO_4$, 3.0; $MgSO_4$, 0.1, pectin, 10.0; and agar, 25.0. The pH of the media was adjusted to 4.5. After incubation, these plates were flooded with CTAB for secondary screening [4,12]. Furthermore, fungal strains were identified by using lactophenol cotton blue stain method to observe their morphology, hyphal characteristics, presence or absence of asexual spores, the arrangement of conidia, and the reproductive structuresunder the bright field microscope under 40X magnification [18,19].

*2.3. Enzyme Production*

2.3.1. Submerged Fermentation for Enzyme Production

Submerged fermentation (SmF) was carried out for the enzyme production. In a conical flask, 50 mL of Hankin's broth was prepared, sterilized and cooled as described previously [20]. Then, 1 mL pre-fermenter inoculum was inoculated. The flasks were incubated in a shaking incubator at 150 rpm and 30 °C for 5–7 days [21]. The pectinase activity in the fermented broth was monitored by using the dinitro salicylic acid (DNS) assay method as described previously [22].Two milliliters of the fermented broth was pipette out into a sterile tube and centrifuged at 8000 rpm for 20 min. The supernatant obtained was used as the crude enzyme and used for analysis. About 1 mL of the crude enzyme and 1 mL of 3% pectin were mixed in a sterile tube and incubated at 50 °C for 15 min. After incubation, 1 mL of the DNS (Dinitrosalicylic acid) reagent was added to stop the hydrolysis reaction. The mixture was then shaken to mix the content and then placed in a boiling water bath for 30 min for color development. The absorbance was then read at 540 nm spectrophotometrically running the enzyme and substrate blanks in parallel. The blank containing 1 mL of 0.5% pectin, 1 mL of sodium acetate buffer (0.1 M, pH 4.2) and 2 mL of DNS reagent was used as a control. One unit of enzyme activity was defined as the amount of enzyme which liberated 1 μmol of galacturonic acid per hour under standard assay conditions.

2.3.2. Effect of Various Factors on Enzyme Production

To study the effect of incubation period, submerged state fermentation was carried out and DNS assay was performed every 24 h until 6 days. The effect of substrate concentration on pectinase production was evaluated using Hankin's broth medium containing different concentrations of pectin (0.5, 1 and 2%) and 1 mL fungal inoculum. The enzyme production was assessed by DNS method after incubating the culture flask at 30 °C for 48 h. To study the effect of temperature on pectinase production, Hankin's broth containing 1% pectin was prepared. Then, 1 mL of the inoculum was

added and incubated separately at 25, 30, and 40 °C for 48 h. After incubation, the enzyme assay was performed using the DNS method.

### 2.3.3. Enzyme Extraction and Partial Purification

The fermentation broth was filtered through Whatman No.1 filter paper and centrifuged (10,000 rpm, 30 min, at 4 °C) for 2–3 times to remove the spores and mycelia of the organism. The filtration process was carried out in a cold condition in ice water to prevent enzyme deactivation. The supernatant represented the soluble crude extract. The crude enzyme was then mixed with three volumes of ice-cold acetone and was allowed to stand for 15 min. The entire content was centrifuged at 4000 rpm at 4 °C for 20 min. The enzyme precipitate was dissolved in a sodium acetate buffer (0.1 M, pH 4.2) and was stored at 4 °C for further use [23].

### 2.4. Characterization of Partially Purified Pectinase Enzyme

#### 2.4.1. Thermal Stability

To assess the thermal stability, the partially purified enzyme extract was heated at 60 °C until 1 h had passed. At 10 min intervals, 1 mL aliquots were withdrawn and mixed with 1 mL of 0.3% (*w/v*) pectin in test tubes and the tubes were incubated at 30 °C for 10 min. The enzyme activity was measured as described by the DNS assay method [24,25].

#### 2.4.2. Effect of Temperature on Enzyme Activity

For temperature assessment, 1 mL of enzyme suspended on sodium acetate buffer (0.1 M, pH 4.2) was added to 1 mL of 0.3% (*w/v*) pectin solution in sterile tubes. The tubes were incubated from 25 to 55 °C for 15 min. After incubation, 2 mL of the DNS reagent was added and incubated in a boiling water bath for 15 min. The enzyme activity was monitored by the DNS assay method.

#### 2.4.3. Effect of pH on Enzyme Activity

The effect of pH on enzyme activity was tested using sodium acetate buffer (0.1 M; pH range, 3.2–5.8), sodium phosphate buffer (0.1 M; pH range, 5.9–7.1) and Tris-HCl buffer (0.1 M; pH range, 7.2–9.0). Tubes containing 0.5 mL respective buffers were mixed with 0.5 mL of the enzyme. Then, 1 mL of 0.3% (*w/v*) pectin solution was added and all the tubes were incubated at 30 °C for 10 min. Afterward, 2 mL of the DNS reagent was added and incubated in boiling water for 15 min. The enzyme activity was measured by DNS assay method.

#### 2.4.4. Effect of Substrate Concentration on Enzyme Activity

Pectin substrates of different concentrations (0.5%, 1.0%, 1.5% and 2.0%) were prepared in different tubes to evaluate the effect of substrate concentration on enzyme activity. One mL of enzyme suspended in acetate buffer (0.1 M, pH 4.2) was mixed with 1 mL of the respective substrate concentrations. Determination of the pectinase activity was done by the DNS assay method.

### 2.5. Application of Pectinase Enzyme

Mature ripened orange fruits (*Citrus sinensis*) were commercially purchased from a local grocery store and stored in food processing laboratory. Prior to juice extraction, the fruits were sorted, washed and peeled. The juice was extracted using sterile small-scale juice extractor available in the laboratory. The extracted juice was pasteurized at 85 °C for 3 min to inactivate the natural fruit enzymes and cooled down to 4 °C. Then, different conical flasks containing 20 mL of orange juice were added with varying concentrations of the enzyme (0%, 0.25%, 0.5%**,** 0.75% and 1%). All the flasks were incubated at 30 °C for 1 h. After incubation, the samples were then heated at 85 °C for 3 min to inactivate the enzyme. The juice treated with different enzyme concentrations was filtered using Whatman No. 1 filter paper. The volume of fruit juice obtained was measured using a 50 mL

measuring cylinder. The total soluble solids (TSS) was determined using a Brix refractometer (model no. CTL-REFM-BR32, LW Scientific) and expressed as degree Brix (°Brix). For clarity, the juice was shaken and 10 mL of juice was centrifuged at 5000 rpm for 10 min. The supernatant portion of the juice was used to determine percent transmittance and absorbance at 540 nm by spectrophotometer (SL−150, Elico) [26].

### 2.6. Statistical Analysis

All the data were statistically analyzed using Office Excel 2019 and OriginPro 8.5 software. The statistically significant (*p*-value < 0.05) of various factors on enzyme activity was determined using one-way ANOVA.

## 3. Results and Discussion

### 3.1. Isolation, Screening, and Identification of the Fungal Isolates

A total of 55 fungal strains were isolated and screened primarily. Based on the zone of hydrolysis, only 14 strains were subjected for secondary screening. During secondary screening, 4 strains designated as Gm, Lco, C, and T showed the highest pectinolytic activity with the zone of hydrolysis of 35, 32, 25, and 15 mm in diameter, respectively. These four strains were morphologically and culturally identified as *Aspergillus* spp. Gm, *Penicillium* spp. Lco, *Fusarium* spp. C, and *Aspergillus* spp. T. Based on primary and secondary screening, *Aspergillus* spp. Gm was found to be a potent strain and hence was selected for further study. The Photograph 1A shows the zone of hydrolysis observed during primary and secondary screening for enzyme production. Similarly, Figure 1B,C show the colony morphology and lactophenol cotton blue staining results of the isolate, respectively. The strains of *Aspergillus* spp. are promising sources of pectinase enzyme. Several previous studies have demonstrated the significance of *Aspergillus* spp. in the commercial production of pectinase enzyme [12,25].

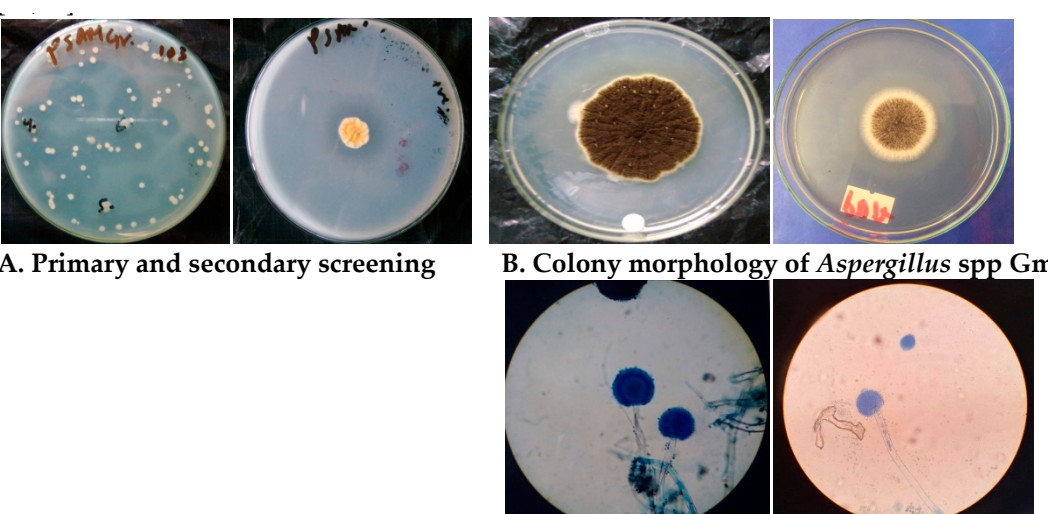

**A. Primary and secondary screening**   **B. Colony morphology of *Aspergillus* spp Gm**

**C. Microscopy of *Aspergillus* spp Gm**

**Figure 1.** (**A**) Primary and secondary screening for *Aspergillus* spp. Gm. (**B**) Colony morphology of *Aspergillus* spp. Gm. (**C**) Lactophenol cotton staining of *Aspergillus* spp. Gm.

### 3.2. Optimization of the Fermentation Conditions for Maximum Enzyme Production

The strain *Aspergillus* spp. Gm was used to study the optimum conditions for maximum enzyme production. This strain showed the highest (106.7 U/mL) pectinase production at 48 h of incubation period. After 48 h, the enzyme production was found to be decreased (Figure 2A). Various other studies have reported the highest enzyme activity of *Aspergillus niger* at 48–72 h, after which the enzyme production was found to be declined [27,28]. The cause of decrease in enzyme production

after certain time interval during incubation might be due to the exhaustion of essential supplements and/or accumulation of toxic metabolites in the culture medium [29]. The shorter fermentation period of 48 h could be advantageous for production of pectinase at industrial scale.

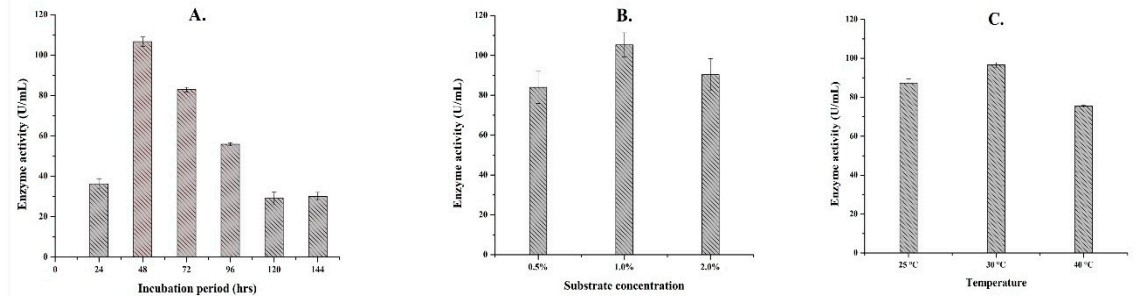

**Figure 2.** Optimization of the fermentation conditions for maximum enzyme production from strain *Aspergillus* spp. Gm. (**A**), effect of incubation period; (**B**), effect of substrate concentration; and (**C**), effect of temperature on enzyme production.

With 1% substrate concentration, strain Gm produces maximum pectinase of 105.2 U/mL. The enzyme production was found to be decreased when the substrate was used more than 1% concentration (Figure 2B). A higher substrate concentration increases the viscosity of culture media and creates nutrient rich environment. Higher nutrients and substrate in the fermentation media inhibit microbial growth lowering enzyme production [30,31]. Previous researches have also stated that lower substrate concentration is effective for enzyme production [32,33]. The requirement of low substrate concentration might be cost effective for large scale production of pectinase enzyme.

At 30 °C, activity of strain Gm was reported to be the highest and produces maximum (96.7 U/mL) pectinase. At a higher temperature, enzyme production declined (Figure 2C). Enzymes are usually denatured at higher temperature resulting decreased activity [34]. Previous studies have reported that the maximum yield of pectinase enzyme from the members of genus *Aspergillus* spp. occur at the temperature range of 30–40 °C [35,36]. Among all the parameters analyzed, only temperature was found to be statistically significant ($p < 0.05$). From these optimization data, this study found that 48 h of incubation period, 1% substrate concentration, and 30 °C temperature could be the optimum cultural conditions for strain Gm to produce maximum pectinase enzyme.

### 3.3. Characteristics of Partially Purified Pectinase

The optimal working ranges of partially purified pectinase enzyme produced by strain Gm were determined based on the assessment of thermal stability, temperature, pH, and substrate concentration. The evaluation of thermal stability indicated that the activity of pectinase reduced approximately by 50% within 10 min of heating at 60 °C. Further heat treatment until 40 min resulted complete loss in the activity of pectinase enzyme (Figure 3A). Thermal stability and activity of pectinases have crucial role in biotechnological processes and food industries. Studies have shown that pectinolytic enzymes can be stable and active at wide range of temperature (30–80 °C) [25,36,37]. The optimum working temperature of pectinase enzyme extracted from strain Gm was found to be 30 °C. At this temperature, pectinase enzyme showed the highest activity of 75.4 U/mL (Figure 3B). There was a significant difference in the activity of pectinase enzyme with various thermal treatment and temperature ($p < 0.05$). This feature of pectinase enzyme possesses great industrial value as it makes the enzyme less susceptible to thermal inactivation. High temperature-tolerant pectinase enzymes have been previously extracted from other various members of *Aspergillus* [29,36,38].

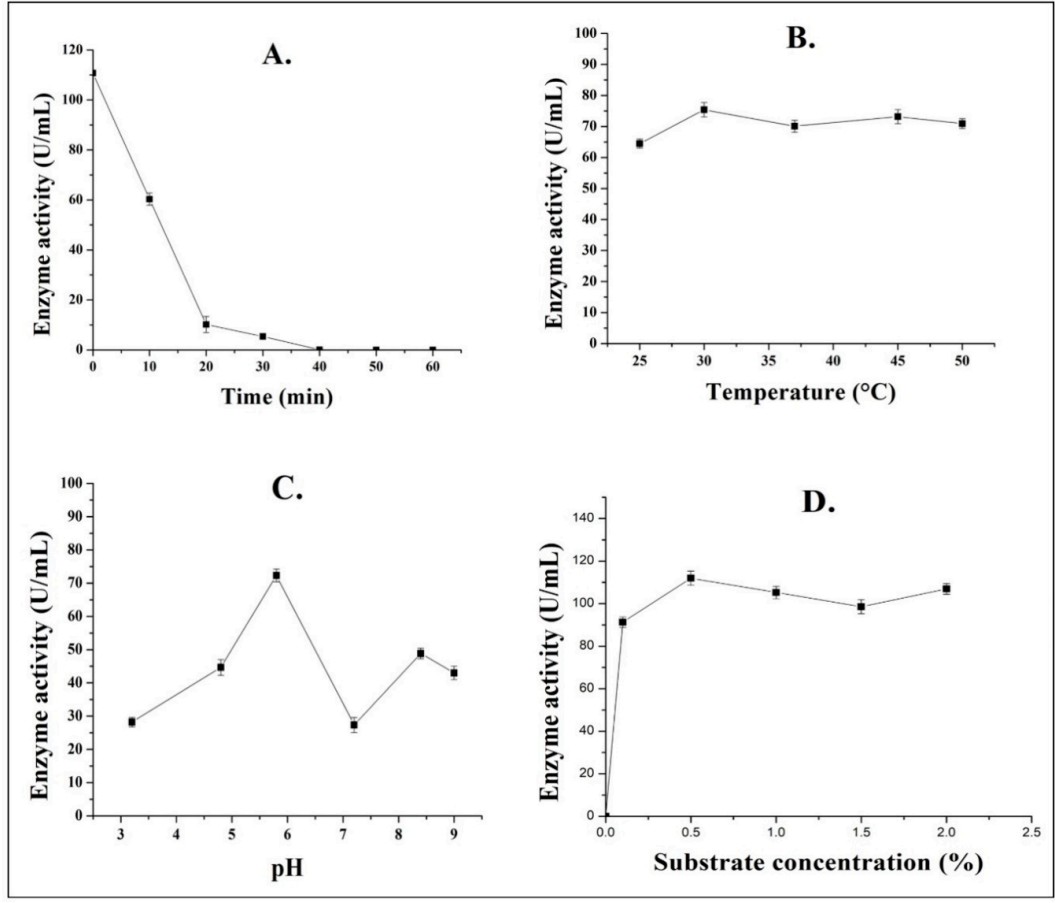

**Figure 3.** Effect of various factors on enzyme activity of partially purified pectinase enzyme from strain *Aspergillus* spp. Gm. (**A**), thermal stability at 60 °C; (**B**), temperature; (**C**), pH; and (**D**), substrate concentration.

The enzyme produced from the strain Gm was found to be active over a wide range of pH (3.2–9.0). However, the optimum enzyme activity (72.3 U/mL) was observed at pH 5.8, indicating that the pectinase enzyme showed higher activity in theslightly acidic range (Figure 3C). The enzyme activity was significantly different with various pH ($p < 0.05$). Most of the studies have reported that pectinolytic enzymes have shown to exhibit higherenzymatic activity at pH range of 4.0–7.0 [25,39,40]. Although the enzyme showed highest activity at pH 5.8, a significant peak at pH 8.0 was observed. This may be due to enzyme's activity and stability. Fungal pectinases are found to be stable from acidic to alkaline pH ranges (4.0–8.0) but their activity and stability may have different peaks at different pH ranges [41]. In a similar study, they observed the stability and optimum activity of fungal pectinases at different ranges, i.e., stability pH 4.0, whereas the optimum activity was found at pH 10.5–11 [42].

Acidic pectinase is useful in preparation of plant protoplast, to perform partial saccharification of sugars, to study phytopathogens invasion in plant, and in other various plant biotechnology applications [15]. However, the activity of pectinase enzyme was observed highest (112.0 U/mL) at 0.5% of substrate concentration. Numerically, the activity of pectinase varies with various substrate concentration (Figure 3D). However, statistically, there was no significant difference in the activity of enzyme with various substrate concentration ($p < 0.05$). Studies have reported that low substrate concentrations are useful to achieve higher enzyme activity [33,43]. High substrate (pectin) concentrations decrease the availability of enzymes and lower the amount of free water in the system. Free water availability is an important factor for maximum enzymatic activity [44]. Therefore, low substrate concentration (0.5%), as observed in this study, could be useful to achieve highest activity of pectinase enzyme.

### 3.4. Application of Pectinase Enzyme in Juice Clarification

The pectinase enzyme extracted from strain Gm was applied for juice clarification. The juice clarification experiments observed that as the concentration of enzyme increased, the yield of the juice, TSS, absorbance value, and % transmittance also increased. At 1% enzyme concentration, the yield of the orange juice, TSS, absorbance value, and % transmittance was found to be the highest, indicating its potential application in juice processing industries (Table 1). There are several studies which have reported that rise in enzyme concentration markedly improved the juice clarification [45,46].

**Table 1.** Effect of different pectinase concentrations on yield, total soluble solids, and transmittance.

| Enzyme Concentration (%) | Yield | | Total Soluble Solids (TSS) | | Absorbance at 540 nm | % Transmittance |
|---|---|---|---|---|---|---|
| | Initial Volume (mL) | Final Volume (mL) | Initial (°Brix) | Final (°Brix) | | |
| 0 | 20.0 | 20.0 | 6.0 | 6.0 | 0.0 | 10.0 |
| 0.25 | 20.0 | 20.2 | 5.0 | 5.5 | 0.48 | 13.0 |
| 0.5 | 20.0 | 20.5 | 5.0 | 5.5 | 0.66 | 17.9 |
| 0.75 | 20.0 | 21.0 | 5.0 | 6.0 | 0.70 | 19.6 |
| 1.0 | 20.0 | 21.7 | 5.0 | 7.0 | 0.75 | 21.0 |

The major impediments to exploit the commercial potential of pectinases are the yield, stability and the cost of enzyme production. In order to obtain high and commercially viable yields of pectinase enzymes, it is essential to optimize the fermentation conditions used for fungal growth and enzyme production. Optimal parameters of the pectinases enzyme biosynthesis from microbial origin varied greatly with the variation of the strains, environmental parameters, and nutritional conditions. Furthermore, the economic feasibility of the microbial enzymes production and its application generally depends on the cost of its production processes [43].

## 4. Conclusions

In conclusion, *Aspergillus* spp. Gm was identified as the promising strain for the production of pectinase enzyme via submerged fermentation. The maximum pectinase production was obtained under optimal cultivation conditions at 48 h of incubation period, 1% substrate concentration, and 30 °C temperature. The pectinase enzyme produced from strain *Aspergillus* spp. Gm showed high thermal stability and reported maximum enzymatic activity at 30 °C temperature, pH 5.8, and 0.5% substrate concentration. In addition, the produced pectinase enzyme in this study showed proven ability in juice clarification, indicating a potential use in food industries. However, further studies must be performed to identify the strain in genetic levels and to assure its commercial application in large-scale food formulation and processing which will be the topic of interest for our research group.

**Author Contributions:** S.K. (Sudeep KC), D.R.J., and B.L.: Conceived and designed the work, performed the experimental work, and collection of the samples. S.K. (Santosh Khanal), T.R.B., B.R.P. and R.D.: Provide analytical tools and contribute in data analysis. D.K.C., and J.U.: data curation, manuscript preparation and manuscript editing. N.K., and V.R.: Contribute in the overall conduct, data analysis, result presentation and submission. All authors have read and agreed to the published version of the manuscript.

**Funding:** This research received no external funding.

**Acknowledgments:** We acknowledge for all the helping hands who helped to conduct this study.

**Conflicts of Interest:** The authors declare that they have no conflict of interest.

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
