# Peer review of "Production, Characterization, and Industrial Application of Pectinase Enzyme Isolated from Fungal Strains"

_fermentation, doi:10.3390/fermentation6020059_

Round 1

Reviewer 1 Report

The article entitled: „Production, characterization, and industrial application of pectinase enzyme isolated from fungal strains“ describes the way of sourcing new fungal isolates capable of producing highly active pectinase.

The strong points of this manuscript are: undertaken research are is still actual and important from the industry point of view so the topic of presened study is interesting for readers. Discussion is quite well described, authors putted many reference sources to discuss obtained results. I like that authors checked the possibility of practical using of isolated enzyme.

The weaknesses of the article are: lack of genetic identification of isolates, basic methods used to assessment of enzyme activity, authors used only one strain for further analysis. So de facto presented results are connected only with one strain, no comparison was made.

*Authors checked the possibility of using obtained enzyme to clarify the home-made orange juice (L 152-153, „The juice was extracted using a domestic juice extractor“) so in my opinion writing „industrial application“ in title is an exaggeration.

*This manuscript present de facto characteristic of enzyme isolated from one strain, characteristic of other strains have been finish on Petri dishes stage so I don’t thing that title should includes „...from fungal strains“.

*L54, „It is recommended to use combination of different kind of pectinases...“, please add WHERE it’s recommended? Specify in which branches of industry/types of technological processes.

*L55, „...in various raw materials..“ please add „such as, and provide some examples of this raw materials“.

*L56-58, the last two sentences of this paragraph sound a little strange. I do not see strong connection of this two with the text presented above them.

*L59-60, Authors say that „ Studies have reported that pectinase accounts for 25% of total industrial enzymes produced in the global market and their economic value is increasing continuously [8].“ But in reference [8] we can find a sentence „Microbial pectinases account for 25% of the global food and industrial enzyme sales and their market is increasing day by day“. I thing that authors interpreted this information in not exactly straight way, because PRODUCE it’s not meant SALES and ECONOMIC VALUE not exactly means MARKET, please rethink.

*L60-61, I don’t think that „fungi are promising sources of several enzymes“, they just are sources of enzymes, but please treat this comment like a soft suggestion.

*L73, on the end of this paragraph it would be more than great if authors could provide the global amount of pectins production as well as some data which show global market of this enzymes, including costs.

*L91, there is a typing error in „identidication“

*Section 2.3., what was the diameter of each colonies and how many colonies were observed on one plate? What was the criteria of dilution the spores? How did you inoculate the spores to obtain a proper amount of colonies? What method of spore counting have you used? Do you think that criteria based on microscope observation are good enough to distinguish species among themselves? There are lack of details, such as type of used microscope, preparation technique description, microscope magnification is even missing.

*Section 2.3.1. Please explain why you choose this old and very basic method with DNS to evaluate activity of pectinase? The method with DNS is using to basic evaluation of reducing sugar concentration but I think that now, we have different, modern and more specific methods to detect enzyme activity. Authors putted ref. [21] as a source of methods information. This publication becomes from 1959 year and it is still according with reducing sugars detection. Maybe authors added DNS to stop the reaction? It is not clear for me... This methods are described very poorly. Please provide the information how many multiplication was every experiment provided.

I can recommend for example this article (you can also find other articles), please rethink your way of describing this section:

Pectinase Activity Determination: An Early Deceleration in the Release of Reducing Sugars Throws a Spanner in the Works! By Alessandra Biz , Fernanda Cardoso Farias, Francine Aline Motter, Diogo Henrique de Paula, Peter Richard, Nadia Krieger, David Alexander Mitchell.

*L 173-175, in my opinion „morphological and cultural“ identification of isolates is definitely not sufficient to say that strains were IDENTIFIED AS... Genetic identification is highly required in this matter.

*Fig1. It is highly recommend to put figures in better quality. The title of this picture should be more precise as authors said that the picture presents A. niger Gm.

*L184, „Various other studies have reported the highest enzyme activity...“, please provide the specific numbers/values of this activity from citing references.

*Fig. 2 should be smaller definitely, there is no need to present obtain data on so huge and bold bar graphs.

*Please complete the discussion with explanation the differences in enzyme activity between non purified and partially-purified enzyme according to results presented in fig.2 B, C and Fig. 3 B, D. Please use another authors results to discuss the differences.

Author Response

Reviewer 1:

The article entitled: “Production, characterization, and industrial application of pectinase enzyme isolated from fungal strains“ describes the way of sourcing new fungal isolates capable of producing highly active pectinase.

The strong points of this manuscript are: undertaken research are is still actual and important from the industry point of view so the topic of presented study is interesting for readers. Discussion is quite well described, authors putted many reference sources to discuss obtained results. I like that authors checked the possibility of practical using of isolated enzyme.

Thank you honorable reviewer for the valuable comments.

The weaknesses of the article are: lack of genetic identification of isolates, basic methods used to assessment of enzyme activity, authors used only one strain for further analysis. So de facto presented results are connected only with one strain, no comparison was made.

Answer to comments: We would like to thank our respected reviewer for the comments. We have done primary screening and secondary screening for the selection of potent enzyme producer. With the limited resources present in the lab, we were not able to do the genetic identification of the isolates. This could be the future research interest as well. Thank you for your valuable suggestions.

*Authors checked the possibility of using obtained enzyme to clarify the home-made orange juice (L 152-153, „The juice was extracted using a domestic juice extractor“) so in my opinion writing „industrial application“ in title is an exaggeration

Answer to comments:  The comment has been addressed in the paper marked in red. Thank you for your valuable suggestions.

*This manuscript present de facto characteristic of enzyme isolated from one strain, characteristic of other strains have been finish on Petri dishes stage so I don’t think that title should include „...from fungal strains“.

Answer to comments: Initially we were looking for both bacteria and fungi, but our thesis work has to be completed in limited time period and hence our advisor designed the project for isolation of fungi only. Thank you for your valuable suggestions.

*L54, “It is recommended to use combination of different kind of pectinases...“, please add WHERE it’s recommended? Specify in which branches of industry/types of technological processes.

Answer to comments: The comment has been addressed in the paper. Thank you for your valuable suggestions.

*L55, “...in various raw materials..“ please add „such as, and provide some examples of this raw materials“. *L56-58, the last two sentences of this paragraph sound a little strange. I do not see strong connection of this two with the text presented above them.

The comment has been addressed in the paper. Thank you for your valuable suggestions.

L59-60, Authors say that „ Studies have reported that pectinase accounts for 25% of total industrial enzymes produced in the global market and their economic value is increasing continuously [8].“ But in reference [8] we can find a sentence „Microbial pectinases account for 25% of the global food and industrial enzyme sales and their market is increasing day by day“. I thing that authors interpreted this information in not exactly straight way, because PRODUCE it’s not meant SALES and ECONOMIC VALUE not exactly means MARKET, please rethink.

The comment has been addressed and new figure regarding the sale has been included in the paper.

*L60-61, I don’t think that „fungi are promising sources of several enzymes“, they just are sources of enzymes, but please treat this comment like a soft suggestion.

The comment has been addressed in the paper.

*L73, on the end of this paragraph it would be more than great if authors could provide the global amount of pectins production as well as some data which show global market of this enzymes, including costs.

The comment has been addressed in the paper.

*L91, there is a typing error in „identidication“.

Spelling has been corrected in the paper. Thank you for your valuable suggestions.

*Section 2.3., what was the diameter of each colonies and how many colonies were observed on one plate? What was the criteria of dilution the spores? How did you inoculate the spores to obtain a proper amount of colonies? What method of spore counting have you used? Do you think that criteria based on microscope observation are good enough to distinguish species among themselves? There are lack of details, such as type of used microscope, preparation technique description, microscope magnification is even missing. *Section 2.3.1. Please explain why you choose this old and very basic method with DNS to evaluate activity of pectinase? The method with DNS is using to basic evaluation of reducing sugar concentration but I think that now, we have different, modern and more specific methods to detect enzyme activity. Authors putted ref. [21] as a source of methods information. This publication becomes from 1959 year and it is still according with reducing sugars detection. Maybe authors added DNS to stop the reaction? It is not clear for me... This methods are described very poorly. Please provide the information how many multiplication was every experiment provided. I can recommend for example this article (you can also find other articles), please rethink your way of describing this section: Pectinase Activity Determination: An Early Deceleration in the Release of Reducing Sugars Throws a Spanner in the Works! By Alessandra Biz , Fernanda Cardoso Farias, Francine AlineMotter, Diogo Henrique de Paula, Peter Richard, Nadia Krieger, David Alexander Mitchell.

The detail of the method exactly used in the work has been included in the paper. The comment has been addressed in the paper.

*L 173-175, in my opinion „morphological and cultural“ identification of isolates is definitely not sufficient to say that strains were IDENTIFIED AS... Genetic identification is highly required in this matter.

Yes, I completely agree with the comment for identification, but unfortunately at the time of work, we had no facility to perform genetic identification in the institution, hence we had to go for conventional method of identification.

*Fig1. It is highly recommend to put figures in better quality. The title of this picture should be more precise as authors said that the picture presents A. niger Gm.

The comment has been addressed in the paper.

*L184, „Various other studies have reported the highest enzyme activity...“, please provide the specific numbers/values of this activity from citing references.

The comment has been addressed in the paper.

*Fig. 2 should be smaller definitely, there is no need to present obtain data on so huge and bold bar graphs.

The comment has been addressed in the paper.

*Please complete the discussion with explanation the differences in enzyme activity between non purified and partially-purified enzyme according to results presented in fig.2 B, C and Fig. 3 B, D. Please use another authors results to discuss the differences.

The comment has been addressed in the paper.

Reviewer 2 Report

Reviewer comment

This paper is interesting to show that the pectinase high-production strain of fungi is screened and its pectinase productivity is assayed. However, the problems of this manuscript are present, described as follows.

  1. Why authors are selected a fungal strain but not the other microorganisms, such as Bacillus of another bacteria? The reason of it is not clear in this manuscript.

  1. Why authors do not identify the screened strain by the molecular identification, for example, using a region of ribosomal DNA?

  1. In this paper, the pectin solution is used in the fungal screening and the assay for pectinase activity. However, it is not shown what type of pectin is used in that experiment. For example, the character of citrus pectin is different from apple pectin. Using the different type of pectin, pectinases produced by the selected fungi is different and whether this type of pectinase is suitable to the subject of this manuscript.

  1. In Fig. 2, there is a mysterious data. Theoretically, enzyme activity of 1.0% substrate in Fig. 2-B is probably same value as the enzyme activity of 30 degree in Fig. 2-C (may be that of 48 h in Fig. 2-A). However, the activity of 1.0% substrate is abnormally higher than that of 30 degree in Fig. 2-C. I think authors need to comment for that data.

  1. The effect of pH on the activity is shown in Fig 3-C. However, using buffer is unknown form pH 4.8 to 7.2, because using buffers are sodium acetate buffer, pH 3.2-4.8 (line 142) and Tris-HCl buffer, pH 7.2-9.0 (line 142). Is this simply mistake? And authors do not also discussion the enzyme activity of pH 7.2-9.0, because “...enzymes have prominent activity at pH range of 4.0-7.0 in the sentence of line 230. Showing Fig. 3-C, the activity in pH 8.0 is higher than that in pH 7.0 and 9.0. In enzymology, the maximum activity of this enzyme is probably pH 8.0 by considering the inhibition by buffer change (Tris-HCl). But author describe productivity of pectinase. Thus, the optimum condition of enzyme productivity is pH 6.0. I think that the discussion of this manuscript is need to change with the consideration of those facts.

I hope authors make well this manuscript by considering above opinions.

Author Response

Reviewer 2:

This paper is interesting to show that the pectinase high-production strain of fungi is screened and its pectinase productivity is assayed. However, the problems of this manuscript are present, described as follows.

Why authors are selected a fungal strain but not the other microorganisms, such as Bacillus of another bacteria? The reason of it is not clear in this manuscript.

Initially we were looking for both bacteria and fungi, but our thesis work has to be completed in limited time period and hence our advisor designed the project for isolation of fungi only.

Why authors do not identify the screened strain by the molecular identification, for example, using a region of ribosomal DNA? .

Yes, I completely agree with the comment for identification, but unfortunately at the time of work, we had no facility to perform genetic identification in the institution, hence we had to go for conventional method of identification.

In this paper, the pectin solution is used in the fungal screening and the assay for pectinase activity. However, it is not shown what type of pectin is used in that experiment. For example, the character of citrus pectin is different from apple pectin. Using the different type of pectin, pectinases produced by the selected fungi is different and whether this type of pectinase is suitable to the subject of this manuscript. Yes, the pectin that we used was citrus pectin, which has been addressed in the paper.

In Fig. 2, there is a mysterious data. Theoretically, enzyme activity of 1.0% substrate in Fig. 2-B is probably same value as the enzyme activity of 30 degree in Fig. 2-C (may be that of 48 h in Fig. 2-A). However, the activity of 1.0% substrate is abnormally higher than that of 30 degree in Fig. 2-C. I think authors need to comment for that data.

Yes, I completely agree with you so that we reanalyzed the raw data obtained during experiment and we found the values which are include in result and discussion part (line 209 and 216) and new Fig. 2 is include in result and discussion part. Here, we got the result that, during the fermentation the best parameter criteria for the production of pectinase using A. niger Gm was at 1% substrate concentration, at 30 degree in 48 hours.

The effect of pH on the activity is shown in Fig 3-C. However, using buffer is unknown form pH 4.8 to 7.2, because using buffers are sodium acetate buffer, pH 3.2-4.8 (line 142) and Tris-HCl buffer, pH 7.2-9.0 (line 142). Is this simply mistake?

Sorry, this was a typographical error which has been corrected. sodium phosphate buffer was used.  

And authors do not also discussion the enzyme activity of pH 7.2-9.0, because “...enzymes have prominent activity at pH range of 4.0-7.0 in the sentence of line 230. Showing Fig. 3-C, the activity in pH 8.0 is higher than that in pH 7.0 and 9.0. In enzymology, the maximum activity of this enzyme is probably pH 8.0 by considering the inhibition by buffer change (Tris-HCl). But author describe productivity of pectinase. Thus, the optimum condition of enzyme productivity is pH 6.0. I think that the discussion of this manuscript is need to change with the consideration of those facts.

Addressed in the results section. Thank you for your valuable comments.

We would like to thank all our honoroable reviewers and editors for their valuable suggestions and comments to bring the best of our manuscript.

Thank you

Round 2

Reviewer 1 Report

Dear authors,

Thank you for provide some of my suggestions into the text. Authors added more details to Material and Method section and correct some typing mistakes, as well as give more numerous data. However there are still few things which should be correct.The quality of figures (for example fig.2) is still unsufficient as well as the way of presented big red block charts. But the main negative aspect of presented article is still the lack of genetically identification so I can't accept that results without strong scientific proofs can be published in high level scientific journals.

Author Response

Reviewer comments

Dear authors

Thank you for providing some of my suggestions into the text. Authors added more details to Material and Method section and correct some typing mistakes, as well as give more numerous data. However there are still few things which should be correct. The quality of figures (for example fig.2) is still unsufficient as well as the way of presented big red block charts. But the main negative aspect of presented article is still the lack of genetically identification so I can't accept that results without strong scientific proofs can be published in high level scientific journals.

Response: Thank you for your valuable comments honorable reviewer.

In our study, strain Gm strain showed the highest (106.7 U/mL) pectinase production at 48 hrs of incubation period. This strain could be used for the industrial production of pectinase. The optimization of fermentation conditions and pectinase enzyme characterization will definitely have a positive impact in the large scale commercial production. To be honest, as we did not have the proper facilities and funds for the genetic identification of the producing strain, we could not have this data and definitely this identification of strain in genetic level will be the interest of our research group in future, provided we get required funds, which we are trying now. We believe this subject should not stop anyone who is interested in commercial production of the pectinase from our strain Gm until and unless the pectinase we characterized is chemically and enzymatically functioning. We are ready to provide this strain Gm to anyone who is either interested to characterize genetically or for commercial production purpose. To be ambiguity and confusion free, we have removed the word niger from the strain Gm.

Finally, Thank you for the suggestions and we highly commend your efforts of pinpointing the mistakes which will certainly add values to this paper. The authors have improved the write-up. All the minor grammatical errors are corrected. The quality of the images presented is improved as suggested. Figure 2 is redrawn and presented in a more clear and accurate form.

Thank you to the honorable editor and reviewer for your time, comments and suggestions.

Dr. Niranjan Koirala

Reviewer 2 Report

Revised manuscript is well than the previous version.  I think to be suitable to publish in Fermentation.

Author Response

Respected reviewer

We, the authors are very much grateful for your comments, suggestions and acceptance of our article for the publication in the Fermentation journal by MDPI.

This has boosted our confidence as a researcher.

Thank you

Round 3

Reviewer 1 Report

Dear Authors,

thank you for making corrections. I still think that the title should be deprived of "industry" element because your results wasn't check in industry circumstances. I also think that this results should be enriched with genetic identification so I suggest to include this part into next research in similar type. Even if you don't have equipment in your lab to do this research, you can always try to start cooperation with other scientific institution and ask for help. I wish you further scientific success.